# Sources of variation and establishment of Russian reference intervals for major hormones and tumor markers

Anna Ruzhanskaya[1], Kiyoshi Ichihara[2]*, Svetlana Evgina[1], Irina Skibo[3], Nina Vybornova[3], Anton Vasiliev[3], Galina Agarkova[1], Vladimir Emanuel[4]

1 Beckman Coulter, Moscow, Russia, 2 Faculty of Health Sciences, Yamaguchi University Graduate School of Medicine, Ube, Japan, 3 Helix Laboratory, Saint-Petersburg, Russia, 4 Pavlov First Saint-Petersburg State Medical University, Saint-Petersburg, Russia

* ichihara@yamaguchi-u.ac.jp

**Data Availability Statement:** All relevant data are within the Supporting information files.

**Funding:** This study was supported by Beckman Coulter, LLC in the form of salaries for AR, SE, and

## Abstract

### Objectives

A multicenter study was organized to explore sources of variation (SVs) of reference values (RVs) for 22 major immunochemistry analytes and to determine reference intervals (RIs) for the Russian population.

### Methods

According to IFCC Committee on Reference Intervals and Decision Limits (C-RIDL) protocol, 758 healthy volunteers were recruited in St. Petersburg, Moscow, and Yekaterinburg. Serum samples were tested for five tumor markers, 17 hormones and related tests by Beckman Coulter's UniCel DxI 800 immunochemistry analyzer. SVs were explored using multiple regression analysis and ANOVA. Standard deviation ratio (SDR) of 0.4 was used as primary guide for partitioning RIs by gender and age.

### Results

SDR for between-city difference was <0.4 for all analytes. Secondary exclusion of individuals was done under the following conditions: for female sex-hormones, those with contraceptives (8%); for CA19-9, those supposed to have negative Lewis blood-group (10.5% males and 11.3% females); for insulin, those with BMI≥28 kg/m² (31%); for the thyroid panel, those with anti-thyroid antibodies (10.3% males; 24.5% females), for CEA those with smoking habit (30% males and 16% females). Gender-specific RIs were required for all analytes except CA19-9, CA15-3, thyroid-related tests, parathyroid hormone, and insulin. Age-specific RIs were required for alpha-fetoprotein, CEA, all sex-hormones for females, FSH and progesterone for both sexes. RIs were generally derived by parametric method after Gaussian transformation using modified Box-Cox formula. Exceptions were growth hormone, estradiol for females in postmenopause, and progesterone for females in premenopause, for which nonparametric method was required due to bimodal distribution and/or insufficient detection limit.

GA, and by Helix Laboratories Services in the form of salaries for IS, NV, and AV. The specific roles of these authors are articulated in the 'author contributions' section. Beckman Coulter, LLC supported the study of the assay reagents and Helix Laboratories Services assisted in the recruitment of volunteers, sample preparations, and provision of sampling equipment. The funders did not have any additional role in the study design, data collection and analysis, decision to publish, or preparation of the manuscript.

**Competing interests:** The authors have read the journal's policy and the authors of this manuscript have the following competing interests: AR, SE, and GA are paid employees of Beckman Coulter, LLC and IS, NV, and AV are paid employees of Helix Laboratories Services. This does not alter our adherence to PLOS ONE policies on sharing data and materials. There are no patents, products in development or marketing products to declare.

**Abbreviations:** 2N-ANOVA, two-level-nested ANOVA; 3N-ANOVA, three-level-nested ANOVA; AACE, American Association of Clinical Endocrinologists; AFP, alpha-fetoprotein; AIT, autoimmune thyroiditis; BC, Beckman Coulter; BMI, body mass index; CA15-3, carbohydrate antigen 15–3; CA125, carbohydrate antigen 125; CA19-9, carbohydrate antigen 19–9; CDL, clinical decision limit; CEA, carcinoembryonic antigen; CI, confidence interval; CLSI, Clinical and Laboratory Standards Institute; C-RIDL, Committee on Reference Intervals and Decision Limits; CV, coefficient of variation; E2, estradiol; EAU, European Association of Urologists; EtOH, ethanol; F, female; FSH, follicle-stimulating hormone; FT3, free triiodothyronine; FT4, free thyroxin; GH, growth hormone; IFCC, International Federation of Clinical Chemistry and Laboratory Medicine; IFU, instruction for use; ISSAM, The International Society for the Study of the Aging Male; ISSM, International Society for Sexual Medicine; LH, luteinizing hormone; LL, lower limit; M, male; Me, median; MP, menopause; MRA, multiple regression analysis; NACB, National Academy of Clinical Biochemistry; OC, oral contraceptives; P, parametric; PRL, prolactin; Prog, progesterone; PSA, prostate-specific antigen; PTH, intact parathyroid hormone; RI, reference interval; $r_p$, standardized partial regression coefficient; RV, reference values; SD, standard deviation; SDR, standard deviation ratio; SHBG, sex hormone binding globulin; SV, sources of variations; TβhCG, total beta human chorionic gonadotropin; Testo, Testosterone; TgAb, anti-thyroglobulin antibody; TPOAb, anti-thyroperoxidase antibody; TRT, Testosterone

## Conclusion

RIs for major hormones and tumor markers specific for the Russian population were derived based on the up-to-date internationally harmonized protocol by careful consideration of analyte-specific SVs.

## Introduction

Each clinical laboratory is expected to establish its own reference intervals (RIs) as recommended in the IFCC/CLSI guideline (C28-A3) [1], but most laboratories in Russia use RIs provided by the reagent manufacturers. They may not match to the Russian population due to a variety of population-specific factors.

Therefore, we joined the global multicenter study on reference value (RVs) coordinated by the IFCC Committee on Reference Intervals and Decision Limits (C-RIDL) in 2013. We recruited 793 healthy volunteers from three major cities: according to the C-RIDL protocol [2] and analyzed biological features of Russian RVs for 34 commonly tested chemistry analytes [3]. As a result, it was revealed that the derived RIs for most chemistry analytes differed greatly from those shown in the reagent inserts, which underlined the importance of determining country-specific RIs for all laboratory analytes.

However, unlike reports on RIs for chemistry analytes, there are not many reports of well-designed studies conducted for establishing RIs for immunoassay analytes. The only comprehensive report available is the IFCC Asian study conducted in 2008~9 involving 3,500 healthy volunteers [4]. The study revealed clear between-country differences for parathyroid hormone (PTH), adiponectin, folate, and vitamin B12 (VB12), but none for other analytes including most of commonly tested tumor markers and reproductive hormones. Another one is a Saudi Arabia study, conducted as a part of IFCC global multicenter study, where 826 apparently healthy individuals were recruited and RIs for 20 immunoassay analytes including five tumor markers, 12 hormones and three vitamins were derived [5]. There are other RI studies, targeting a smaller number of analytes, such as tumor markers [6, 7] and thyroid hormones [8, 9]. Besides, no comprehensive analyses of biological sources of variations have been performed so far except for a recent report from a Chinese group collaborating in the C-RIDL global study, which established RIs for eight male sex hormone-related analytes and seven thyroid hormones and analyzedtheir SVs [10, 11].

In this second part of our RI study, we targeted 22 major immunoassay analytes. They include five tumor markers, eight reproductive hormones and related tests, five thyroid function tests, and four other hormones. By use of the same statistical methods as for the first part, we tried to establish the RIs specific to the Russian population in careful consideration of SVs of each analyte.

## Materials and methods

### 1) Source data and target analytes

The study protocol, including methods for invitation, provision of information about the study for volunteers, taking informed consent and questionnaire regarding current health status and lifestyle, was approved by the Ethic Committee of City Hospital #40, Saint-Petersburg. The scheme used for recruitment, sampling, and measurements was described in the first part of our report, which dealt with RIs and sources of variation of chemistry analytes [3]. In brief,

replacement therapy; TSH, thyroid-stimulating hormone; TT3, total triiodothyronine; TT4, total thyroxin.

758 healthy volunteers (350, 46% men, 408, 54% women) of 18–65 year old were recruited from three regions: Sankt-Petersburg (North-West region: N = 506, 67%), Moscow (Central region: N = 117, 15%) and Yekaterinburg (Ural region: N = 135, 18%) (S1 Table). They were chosen according to inclusion and exclusion criteria stipulated in the C-RIDL protocol, and blood samples were drawn at basal conditions [2]. The part 2 of the report deals with RVs evaluated for a total of 24 analytes measured by immunoassays: carcinoembryonic antigen (CEA), alpha-fetoprotein (AFP), CA19-9, CA125, CA15-3, insulin, cortisol, testosterone (Testo), sex hormone-binding globulin (SHBG), estradiol, progesterone (Prog), luteinizing hormone (LH), follicle stimulating hormone (FSH), total beta human chorionic gonadotropin (TβhCG), prolactin (PRL), growth hormone (GH), thyroid stimulating hormone (TSH), free thyroxine (FT4), free triiodothyronine (FT3), total thyroxine (TT4), total triiodothyronine (TT3), anti-thyroid peroxidase antibody (TPOAb), anti-thyroglobulin antibody (TgAb) and parathyroid hormone (PTH). All were measured by using the UniCel DxI 800 immunochemistry analyzer (Beckman Coulter Inc., USA) according to manufacturer's assay instructions and requirements. Characteristics of the analytes, including their full name, abbreviation, unit, and assay principle are listed in S2 Table. For TSH and TβhCG, after completion of the measurements, the new assays became available, and thus they were tested again using serum aliquots kept stocked at -80˚C. The new TSH and TβhCG assays are standardized to the new highly purified WHO standards: i.e., 3rd International standard (IS) (81/565) and 5th IS (07/364) respectively; the properties of the previous and current reagents are compared in S3 Table.

## 2) Quality control

Quality control was performed in two ways. One was through twice daily measurement of 2 or 3 levels of QC specimens obtained from Beckman Coulter Inc. and Bio-Rad Laboratories, Inc. The other was through daily measurement of a mini panel composed of six sera from healthy volunteers (3 women and 3 men) as described in the common protocol [2, 13]. Based on repeated mini panel measurements, between-day coefficient of variation (CV) was calculated for each analyte. The CV of any analyte did not exceed the allowable limit based on the criterion described in the protocol (i.e., ½ of $CV_I$: within individual CV, presented in the EFLM website (https://biologicalvariation.eu/meta_calculations) (S2 Table). Additionally, as a part of the study aiming at worldwide comparison of RVs, the panel of 40 sera for immunoassays provided by C-RIDL were measured in four batches over a period of four days.

## 3) Statistical procedures

Data analyses and statistical methods used were those recommended in the C-RIDL protocol [2, 12, 13]. Details are descried in Part 1 of our report [3].

**3–1) Analyses for biological sources of variations.** The multiple regression analysis (MRA) was performed by setting RVs of each analyte as object variable and following factors as explanatory variables: sex, age, body mass index (BMI), the levels (see below) of cigarette smoking, alcohol consumption and regular physical exercise [14]. Standardized partial regression coefficient ($r_p$), which corresponds to the partial correlation coefficient. In reference to the Cohen's guide [15] of "effect size" for correlation coefficient (r): r = 0.1 (small) and r = 0.3 (medium), as a middle point, $|0.2| \leq r_p$ was interpreted as a practically significant factor influencing the reference values. The levels of smoking, alcohol consumption and physical exercises were classified into 3, 5 and 8 categories, respectively, using the following criteria: none, ≤20, > 20 cigarettes/day; none, <12.5, 12.5–25, 25–50, >50 g ethanol/day; none, 1–7 days/week.

**3–2) Criteria for partitioning RVs.** By use of 3-level nested ANOVA, between-sex, between-age, between-city, and between-individual variations were computed each as

standard deviation (SD): SDsex, SDage, SDcity, and SDbtw-indiv (= $SD_G$ by common notation). Relative magnitude of each SD to the $SD_G$ was calculated as SD ratio (SDR): $SDR_{sex}$, $SDR_{age}$, and $SDR_{city}$. After absence of between-city differences was confirmed by the criterion described below, $SDR_{age}$ specific for each sex was calculated by one-way ANOVA. As an additional analysis for analytes with obvious BMI-related changes, SDR for BMI ($SDR_{BMI}$) was computed for each sex by two-level nested ANOVA with age set as a covariate.

The need for partition of RVs was considered by setting SDR≥0.4 as a primary guide [2]. However, SDR may be too sensitive when the width of RI that constitutes the denominator of SDR is narrow. Conversely, SDR may be insensitive when between-subgroup differences occur only at the periphery of distribution (LL or UL) because SDR represents between-subgroup bias at the center of the distributions. Therefore, we additionally considered actual difference (bias) at LL or UL as "bias ratio" (BR) using the following formula illustrated for a case of gender difference:

$$BR_{LL} = \frac{|LL_M - LL_F|}{(UL_{MF} - LL_{MF})/3.92}, \quad BR_{UL} = \frac{|UL_M - UL_F|}{(UL_{MF} - LL_{MF})/3.92}$$

where subscript M, F, and MF represent male, female, and male+female, respectively. The denominator of each formula represents the standard deviation ($SD_{RI}$) comprising the RI, the width of which corresponds to 3.92 times $SD_{RI}$.

In accordance with the convention of allowable bias specification of a minimum level: $0.375 \times SD_G$ (= $SD_{RI}$) [16], we regard $BR_{UL}$>0.375 as an auxiliary threshold for partitioning RVs when SDR does not match to actual between-subgroup difference at ULs (or LLs).

In performing MRA and ANOVA, RVs of analytes that exhibited highly skewed distributions were transformed logarithmically. The corresponding analytes were marked in Table 1. In that case, SDRs were computed by reverse transformation of each SD component that was calculated under the transformed scale as described elsewhere [17].

**3–3) Derivation of reference intervals.** RIs were derived by both parametric and non-parametric methods. The former was performed after normalizing data by use of the modified Box-Cox power transformation formula [18]. The validity of the parametric method was confirmed by the linearly of cumulative distribution of RVs on probability paper plot [13] and by the Kolmogorov-Smirnov test. If the transformation failed, the non-parametric method was used. The 90% confidence interval (CI) of the lower limit (LL) and upper limit (UL) of RI was calculated by the bootstrap method through random resampling of the same dataset 50 times. Accordingly, the final LL and UL of RI both by parametric and nonparametric methods was chosen as the average of iteratively derived LLs and ULs.

# Results

## 1. Sources of variation of RVs

SVs of each analyte were evaluated by MRA and ANOVA as described in the part 1 of this report [3] and respective results are listed in Tables 1 and 2. In the following sections, the findings of 22 laboratory parameters were divided into four groups according to their categories: tumor markers (AFP, CEA, CA19-9, CA125, CA15-3), the reproductive panel (PRL, LH, FSH, TβhCG, estradiol, progesterone, testosterone and SHBG), thyroid function tests (TSH, FT4, FT3, TT4, TT3), and miscellaneous ones (insulin, cortisol, GH and PTH). No apparent between-city differences ($SDR_{city}$) were observed for any analyte with the highest $SDR_{city}$ of 0.21 observed for TSH (data omitted). Therefore, all the data from the three cities were merged in the subsequent analyses.

**Table 1. Results of multiple regression analysis for sources of variations of RVs.**

| | log scale | Male | | | | | | | Female | | | | | | |
|---|---|---|---|---|---|---|---|---|---|---|---|---|---|---|---|
| | | n | R | age | BMI | ExerLvl | SmkLvl | DrkLvl | n | R | age | BMI | ExerLvl | SmkLvl | DrkLvl |
| AFP | ○ | 339 | 0.26 | 0.18 | 0.06 | -0.14 | -0.05 | 0.07 | 396 | 0.38 | **0.33** | 0.06 | -0.07 | 0.02 | -0.02 |
| CEA | ○ | 339 | 0.35 | 0.19 | 0.13 | 0.09 | **0.27** | 0.02 | 396 | 0.36 | **0.30** | 0.04 | 0.04 | 0.20 | 0.03 |
| CA19-9 | ○ | 303 | 0.35 | **0.35** | -0.03 | -0.05 | -0.02 | -0.01 | 350 | 0.11 | 0.02 | -0.01 | 0.05 | -0.08 | 0.06 |
| CA125 | ○ | 338 | 0.19 | 0.02 | 0.18 | 0.00 | 0.00 | -0.02 | 386 | 0.33 | **-0.32** | 0.06 | -0.03 | 0.12 | 0.01 |
| CA15-3 | ○ | 339 | 0.26 | 0.20 | 0.13 | -0.05 | -0.03 | -0.06 | 395 | 0.29 | **0.22** | 0.08 | 0.00 | -0.02 | 0.12 |
| PRL | ○ | 338 | 0.21 | -0.09 | -0.07 | -0.07 | -0.12 | -0.10 | 365 | 0.50 | **-0.44** | -0.05 | -0.04 | -0.15 | 0.13 |
| LH | ○ | 338 | 0.29 | **0.27** | -0.16 | 0.06 | -0.04 | -0.05 | 364 | 0.56 | **0.56** | -0.01 | 0.04 | -0.02 | 0.01 |
| FSH | ○ | 338 | 0.45 | **0.46** | -0.04 | -0.03 | -0.03 | -0.03 | 366 | 0.81 | **0.82** | -0.03 | 0.01 | 0.02 | 0.00 |
| TβhCG | ○ | | | | | | | | 334 | 0.67 | **0.62** | 0.09 | -0.07 | 0.05 | -0.01 |
| Estradiol | ○ | 338 | 0.11 | -0.04 | 0.03 | -0.08 | -0.09 | 0.00 | 366 | 0.62 | **-0.64** | 0.04 | -0.01 | -0.06 | -0.02 |
| Prog | ○ | 338 | 0.36 | **-0.24** | -0.19 | -0.11 | -0.06 | -0.05 | 366 | 0.51 | **-0.46** | -0.09 | -0.06 | 0.05 | 0.00 |
| Testo | ○ | 333 | 0.54 | -0.13 | **-0.48** | 0.06 | 0.07 | -0.05 | 369 | 0.49 | **-0.53** | 0.14 | 0.02 | 0.00 | 0.03 |
| SHBG | ○ | 292 | 0.57 | **0.42** | **-0.45** | 0.11 | 0.19 | -0.01 | | | | | | | |
| TSH | ○ | 305 | 0.09 | -0.03 | -0.05 | 0.04 | -0.03 | 0.01 | 295 | 0.22 | 0.00 | 0.04 | -0.01 | **-0.22** | 0.08 |
| FT4 | | 303 | 0.18 | -0.08 | -0.09 | -0.02 | -0.05 | -0.09 | 292 | 0.11 | 0.06 | -0.07 | -0.01 | 0.07 | 0.05 |
| FT3 | | 302 | 0.27 | -0.21 | 0.18 | -0.06 | 0.06 | -0.06 | 294 | 0.15 | -0.13 | 0.14 | -0.04 | -0.01 | 0.03 |
| TT4 | | 282 | 0.15 | 0.06 | 0.08 | -0.09 | 0.01 | 0.00 | 282 | 0.17 | -0.05 | 0.18 | 0.04 | -0.05 | 0.02 |
| TT3 | | 281 | 0.23 | -0.12 | 0.13 | -0.11 | 0.06 | -0.09 | 281 | 0.21 | **-0.21** | 0.11 | 0.06 | -0.02 | -0.10 |
| Insulin | ○ | 338 | 0.67 | -0.15 | **0.66** | -0.15 | -0.10 | -0.02 | 395 | 0.59 | -0.09 | **0.61** | -0.04 | -0.05 | -0.08 |
| Cortisol | | 339 | 0.26 | -0.18 | -0.12 | -0.11 | -0.08 | 0.04 | 396 | 0.26 | -0.04 | **-0.21** | -0.11 | -0.09 | -0.01 |
| GH | ○ | 339 | 0.38 | **0.31** | **-0.24** | 0.15 | 0.05 | -0.11 | 396 | 0.36 | 0.12 | **-0.38** | 0.06 | 0.04 | 0.03 |
| PTH | ○ | 340 | 0.36 | **0.24** | **0.20** | -0.05 | 0.00 | 0.02 | 396 | 0.38 | 0.17 | **0.24** | 0.02 | -0.08 | -0.08 |

**1–1. Tumor markers.** Sex-related changes with $SDR_{sex} \geq 0.4$ were not observed in any tumor marker as shown in Table 2. However, by close look at S1 Fig for sex- and age-related changes, RVs of CA125 in females are appreciably higher than males until 50 years of age, but lower thereafter with $SDR_{age}$ of 0.35. This unmatched age-related change between the two sexes led to spuriously low $SDR_{sex}$ of 0.25 for CA125. While |BR| for between-sex difference is well above 0.375. Therefore, we chose to partition RVs by sex for CA125.

Based on $|r_p| \geq 0.2$ considered as a practically significant level of association (Table 1), age-related changes in RVs were observed for the following analytes with their $r_p$ shown in the parenthesis: in males, CA19-9 (0.35), and CA15-3 (0.20); in females, AFP (0.33), CA125 (−0.32), CEA (0.30) and CA15-3 (0.22) in the descending order of $|r_p|$.

This female dominant age-related change of AFP and CA125 was clearly seen in Fig 1 and S1 Fig, respectively. However, the magnitude of age-related changes in terms of $SDR_{age}$ was all slightly below 0.4 except that of AFP in females (0.49).

As SVs other than sex and age, smoking habit-related changes in RVs was noted by MRA in CEA as shown in Fig 2. Another important factor as a SV was the Lewis blood group-related change in CA19-9. Although we have not confirmed it by actual analysis of the blood type, S1 Fig clearly showed a distinct cluster of data points below the detection limit of 0.8 KIU/L. Assuming them as representing Lewis negative individuals in Russia, its prevalence among healthy individuals are 10.5% (36/341) in males and 11.3% (45/396) in females. With these observations, we derived RIs for CEA after excluding individuals with smoking habits, and for CA19-9 after excluding individuals with values below the detection limit.

**Table 2. List of SDRs representing between-subgroup variations by sex, age, and BMI.**

| | Analyte | SDRsex | SDRage M | SDRage F | SDR$_{BMI}$ M | SDR$_{BMI}$ F |
|---|---|---|---|---|---|---|
| **Tumor markers** | AFP | 0.00 | 0.29 | **0.49** | | |
| | CEA | 0.23 | 0.19 | **0.36** | | |
| | CA19-9 | 0.19 | **0.36** | 0.08 | | |
| | CA125 | 0.25 | 0.20 | **0.36** | | |
| | CA15-3 | 0.18 | 0.21 | 0.29 | | |
| **Reproductive hormones** | PRL | 0.23 | 0.16 | **0.57** | | |
| | LH | **1.40** | **0.39** | **0.88** | | |
| | FSH | **1.21** | **0.52** | **2.10** | | |
| | TβhCG | | | **1.15** | | |
| | Estradiol | 0.16 | 0.07 | **0.98** | | |
| | Prog | 0.28 | **0.43** | **0.63** | | |
| | Testo | **5.28** | 0.18 | **0.47** | **0.65** | 0.00 |
| | SHBG | | **0.54** | | **0.35** | |
| **Thyroid function tests** | TSH | 0.00 | 0.17 | 0.00 | | |
| | FT4 | 0.07 | 0.10 | 0.00 | | |
| | FT3 | **0.45** | 0.16 | 0.00 | | |
| | TT4 | 0.14 | 0.00 | 0.00 | | |
| | TT3 | 0.03 | 0.00 | 0.16 | | |
| **Other hormones** | Insulin | 0.06 | 0.00 | 0.16 | **0.91** | **0.82** |
| | Cortisol | 0.22 | 0.28 | 0.19 | | |
| | GH | **1.27** | 0.23 | 0.14 | 0.08 | **0.31** |
| | PTH | 0.00 | **0.36** | **0.33** | | |

**1–2. Reproductive panel.** From Fig 1 and S1 Fig as well as from Tables 1 and 2, prominent sex and age-related changes were observed in all eight analytes in the reproductive panel. RVs of estradiol and progesterone in females showed an abrupt reduction at around 50 years of age (a peak time of menopause) with $r_p$ of −0.64 and −0.46 and SDR$_{age}$ of 0.98 and 0.63, respectively. It is notable that postmenopausal values are well below those of males. In contrast, RVs of estradiol in males stay unchanged by age, while RVs of progesterone in males decrease slightly with age (SDR$_{age}$ 0.43). For testosterone, between-sex difference is very prominent with female testosterone levels approximately 1/10th of those of males. Interestingly, age-related reduction of testosterone is more prominent in females. It was shown that testosterone RVs in males were affected by BMI, but not by age ($r_p$: −0.48 and −0.13, respectively). On the contrary, RVs in females were affected by age, but not by BMI ($r_p$: −0.53 and 0.14, respectively). It was confirmed by a change of SDR after exclusion of patients with BMI>28, SDR$_{age}$ was 0.47 in females and 0.18 in males (Fig 1, S1 Fig).

In females, LH, FSH, and TβhCG showed an abrupt surge after menopause with SDR$_{age}$ of 0.88, 2.10, and 1.15, and with $r_p$ of 0.56, 0.82, and 0.62, respectively. On the other hand, in males, the age-related elevation of LH and FSH are slight and gradual with SDR$_{age}$ of 0.39 and 0.52, respectively. For PRL, the reduction by age is shown only in females with SDR$_{age}$ of 0.57.

From these observations and using a criterion of SDR$_{age}$≥0.40, in females, partition of RVs by the status of menopause as self-reported in the questionnaire was essential for PRL, LH, FSH, TβhCG, estradiol, progesterone, and testosterone. For the age-related changes of FSH and progesterone in males, as a boundary value for partition, we chose 45 years of age, as roughly representing a mid-point of changes in RVs with age.

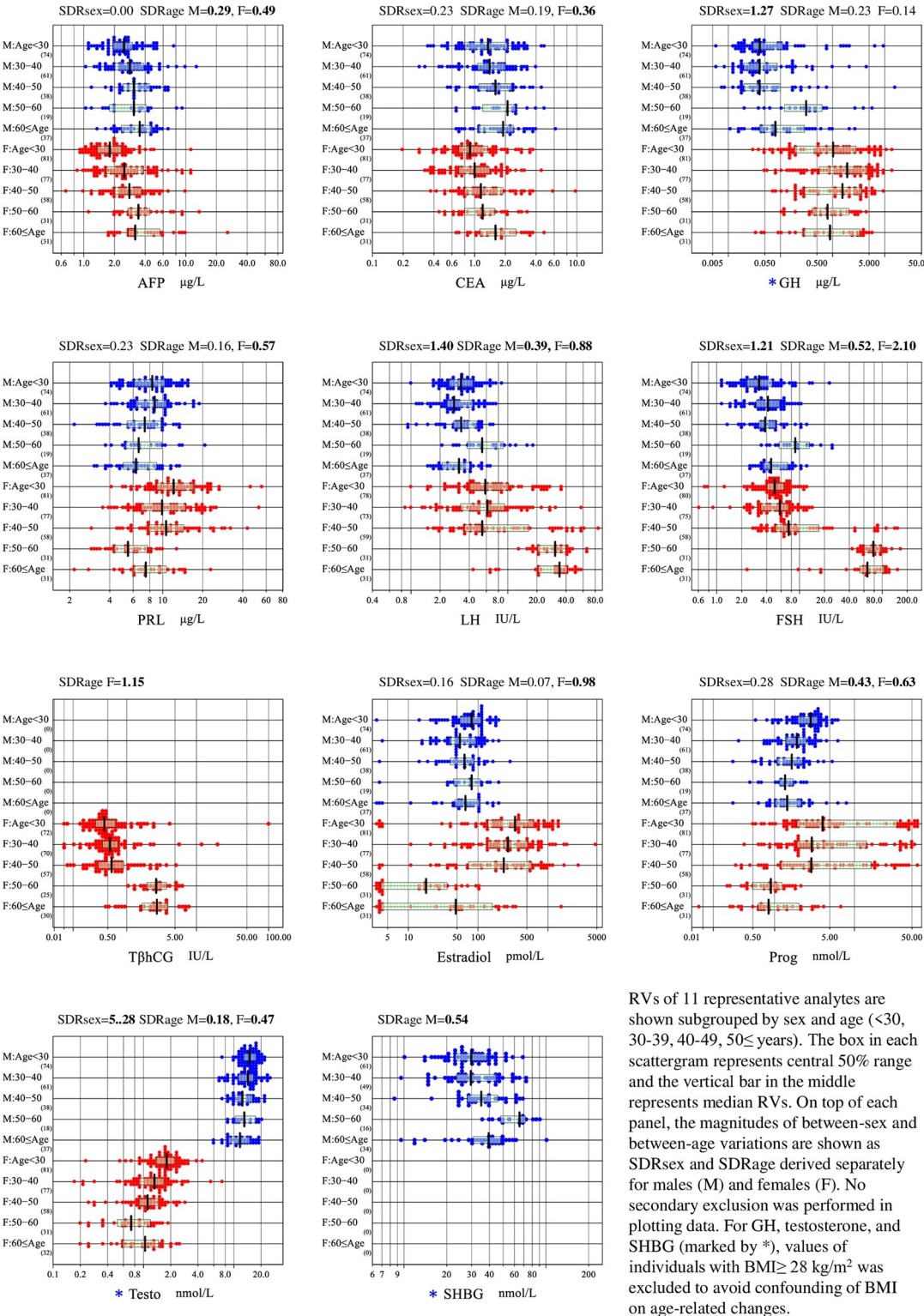

RVs of 11 representative analytes are shown subgrouped by sex and age (<30, 30-39, 40-49, 50≤ years). The box in each scattergram represents central 50% range and the vertical bar in the middle represents median RVs. On top of each panel, the magnitudes of between-sex and between-age variations are shown as SDRsex and SDRage derived separately for males (M) and females (F). No secondary exclusion was performed in plotting data. For GH, testosterone, and SHBG (marked by *), values of individuals with BMI≥ 28 kg/m² was excluded to avoid confounding of BMI on age-related changes.

**Fig 1. Sex and age-related changes in RVs of 11 representative analytes.** RVs of 11 representative analytes are shown subgrouped by sex and age (<30, 30~39, 40~49, 50≤ years). The box in each scattergram represents central 50% range and the vertical bar in the middle represents median RVs. On top of each panel, the magnitudes of between-sex and between-age variations are shown as $SDR_{sex}$ and $SDR_{age}$ derived separately for males (M) and females (F). No secondary exclusion was performed in plotting data. For GH, testosterone, and SHBG (marked by *), values of individuals with BMI≥ 28 kg/m² was excluded to avoid confounding of BMI on age-related changes.

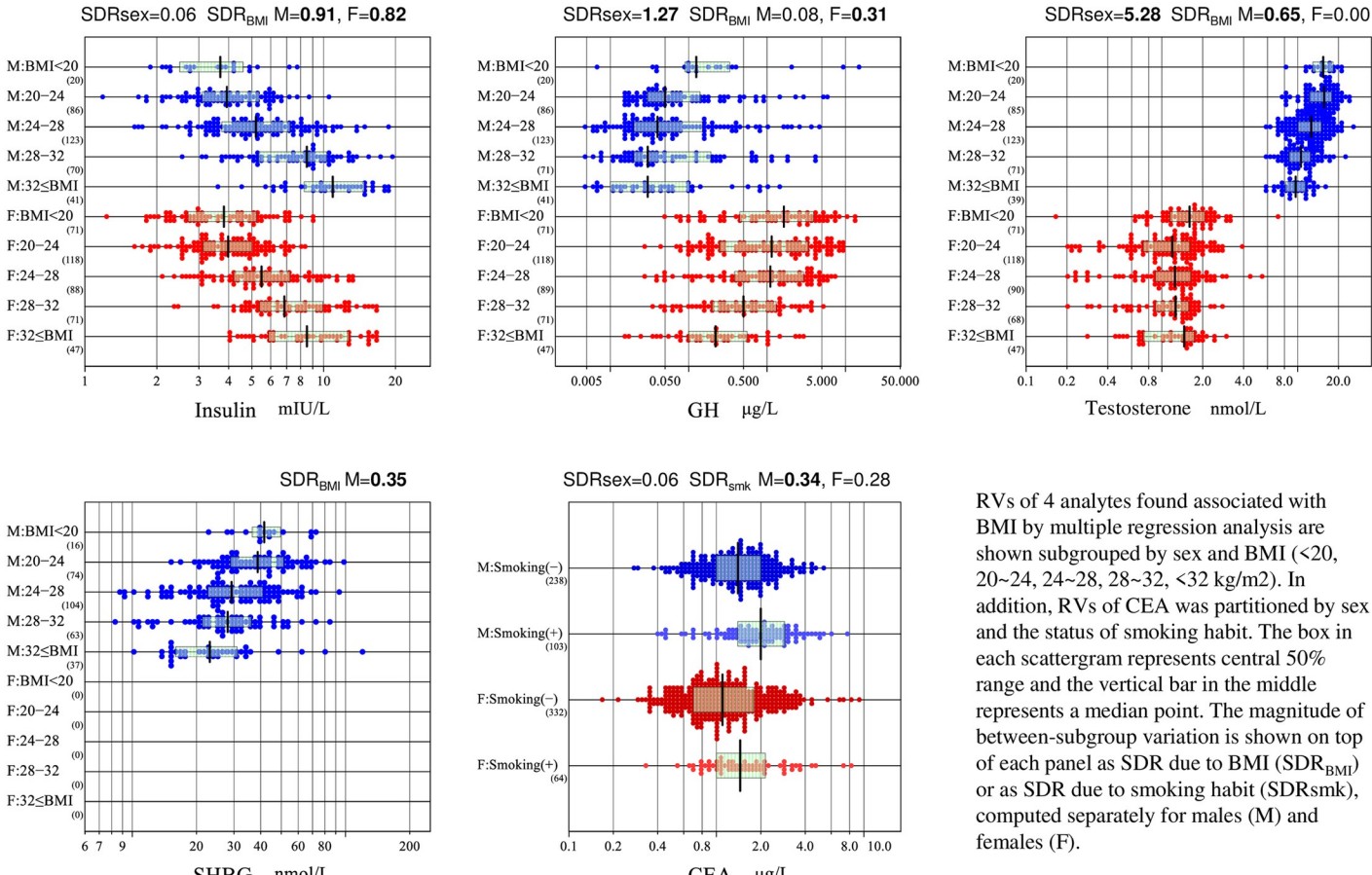

**Fig 2. Association of BMI or smoking habit with RVs of selected analytes.** RVs of 4 analytes found associated with BMI by multiple regression analysis are shown subgrouped by sex and BMI (<20, 20~24, 24~28, 28~32, <32 kg/m²). In addition, RVs of CEA was partitioned by sex and the status of smoking habit. The box in each scattergram represents central 50% range and the vertical bar in the middle represents a median point. The magnitude of between-subgroup variation is shown on top of each panel as SDR due to BMI (SDR$_{BMI}$) or as SDR due to smoking habit (SDR$_{smk}$), computed separately for males (M) and females (F).

Regarding BMI-related changes of the reproductive panel, in addition to testosterone, SHBG of males showed high association with $r_p$ (SDR$_{BMI}$) of −0.45 (0.35) (Tables 1 and 2). The trend is clearly shown in Fig 2. For the two analytes, we examined the effect of excluding individuals with BMI≥28 on their RIs (see below).

**1–3. Thyroid function tests.** In the analyses of thyroid function tests, we first identified cases with subclinical autoimmune thyroiditis (AIT) by use of the criteria of TgAb≥4 KIU/L or TPOAb≥9 KIU/L, which are provided in the kit inserts. Prior to deriving RIs, we found the cutoff values were appropriate as a proximal point of tailing values in the distributions in S1 Fig (in the last two panels). The prevalence of individuals exceeding either of the cutoff values was 10.3% (36/350) in males and 24.5% (100/408) in females. The comparison of five thyroid function test results between individuals with and without the autoantibodies are shown in Fig 3. It is apparent that only RVs of TSH differed between the two groups with SDR of 0.58 (male) and 0.48 (female) for the status of AIT or SDR$_{AIT}$. With the results, in the subsequent analyses including derivation of RIs for all the thyroid function tests, we excluded individuals judged as AIT as well as those under thyroxine replacement therapy.

By MRA, age-related reduction in RVs were observed for FT3 ($r_p$ = −0.21 in males), and TT3 (−0.21 in females). While SDR$_{age}$ was only 0.16 for both tests. Therefore, we chose not to

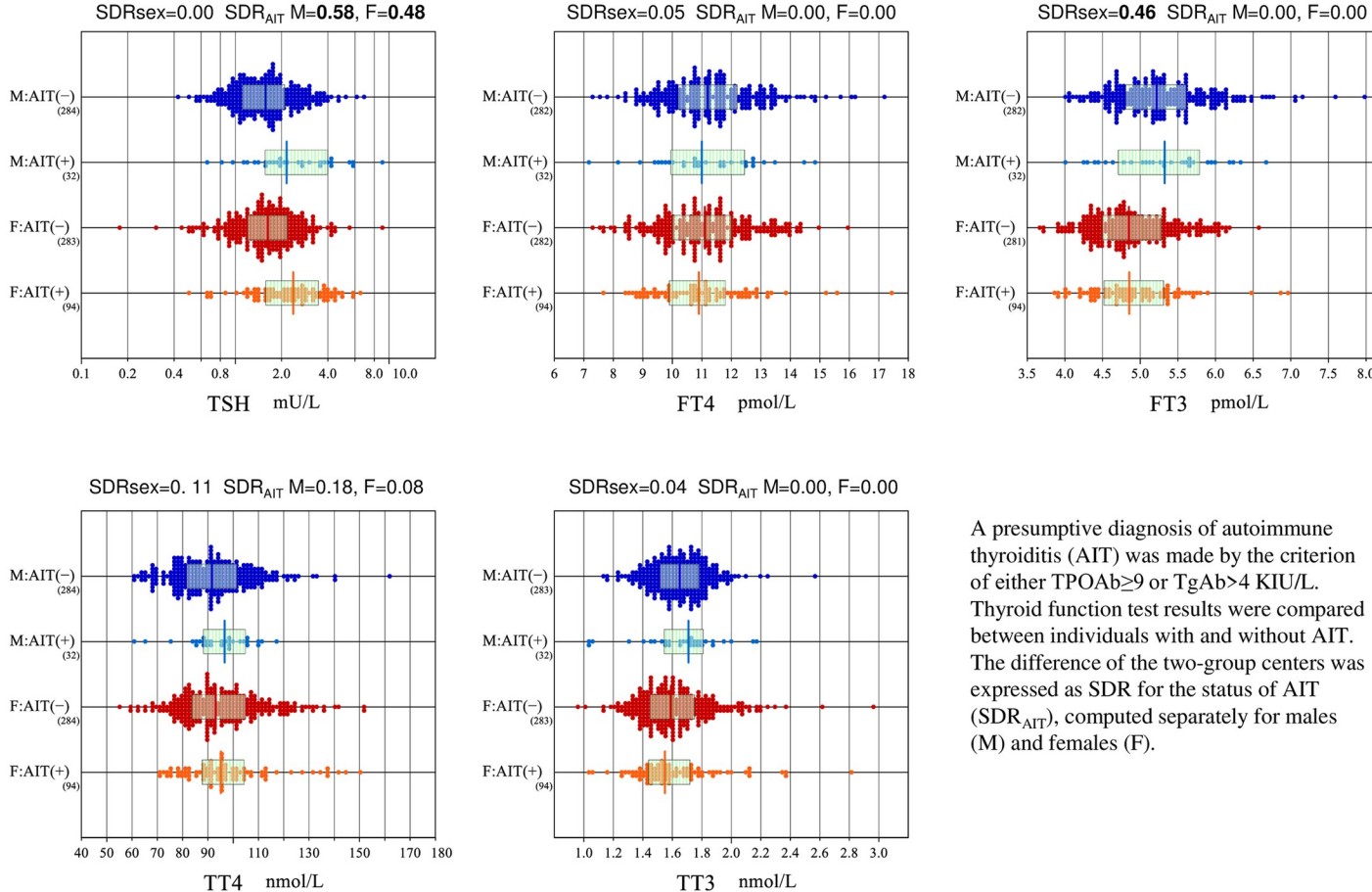

**Fig 3. Influence of autoimmune thyroiditis on thyroid function tests.** A presumptive diagnosis of autoimmune thyroiditis (AIT) was made by the criterion of either TPOAb≥9 or TgAb>4 KIU/L. Thyroid function test results were compared between individuals with and without AIT. The difference of two-group centers was expressed as SDR for the status of AIT ($SDR_{AIT}$), computed separately for males (M) and females (F).

partition RVs by age for any of the thyroid function tests. In fact, S1 Fig showed that age-related changes in RVs of FT3 and TT3 were not conspicuous. As for sex-related change, it was observed only in FT3 with $SDR_{sex}$ of 0.45 (values higher in males) (Table 2).

**1–4. Miscellaneous hormones.** MRA revealed a conspicuous BMI-related increase of insulin and moderate decrease of GH with $r_p$ of 0.66 and –0.24, respectively, in males, and 0.61 and −0.38 in females. These trends are clearly seen in Fig 2, but in terms of $SDR_{BMI}$, only that of insulin showed high values of 0.91 (males) and 0.82 (females) in Table 2. Therefore, we examined the effect of excluding individuals with high BMI as described below in deriving RIs for insulin and GH.

For age-related changes, RVs for GH and PTH in males showed an increase with age ($r_p$ for age: 0.31 and 0.24, respectively) as shown in Table 1. However, in terms of $SDR_{age}$, that of GH is well below 0.4, apparently indicating age-related increase of GH is counter-balanced by BMI-related reduction of GH (i.e., BMI increases with age).

## 2. Derivation of RIs

According to the scheme for partition or secondary exclusion of RVs described above in details, RIs for all the 22 parameters were derived and summarized in Table 3. It is of note that

**Table 3. List of RIs adopted for all analytes with or without partition by sex and age.**

| | | | | Partitioning/exclusion | | | 90%CI of LL | | Reference interval | | | 90%CI of UL | |
|---|---|---|---|---|---|---|---|---|---|---|---|---|---|
| Method | Test Item | Unit | Sex | Age | Exclusion | n | LL-L | LL-H | LL | Me | UL | UL-L | UL-H |
| P | AFP | µg/L | M+F | <45 | | 420 | 0.97 | 1.11 | 1.0 | 2.4 | 7.0 | 5.79 | 8.15 |
| P | | | M+F | ≥45 | | 307 | 1.24 | 1.66 | 1.5 | 3.3 | 8.7 | 7.58 | 9.78 |
| P | CEA | µg/L | M | All | Smoker | 238 | 0.42 | 0.55 | 0.48 | 1.44 | 3.84 | 3.36 | 4.31 |
| P | | | F | <45 | Smoker | 180 | 0.27 | 0.40 | 0.33 | 0.95 | 3.32 | 2.77 | 3.86 |
| P | | | F | ≥45 | Smoker | 151 | 0.43 | 0.52 | 0.47 | 1.35 | 5.19 | 4.02 | 6.35 |
| P | CA19-9 | kIU/L | M+F | All | Extreme low | 639 | 2.0 | 2.6 | 2.3 | 5.7 | 29.3 | 24.9 | 33.7 |
| P | CA125 | kIU/L | M | All | | 340 | 3.6 | 4.3 | 3.9 | 10.0 | 27.5 | 24.4 | 30.6 |
| P | | | F | All | | 392 | 4.3 | 5.5 | 4.9 | 12.4 | 38.7 | 33.0 | 44.4 |
| P | CA15-3 | kIU/L | M+F | All | | 728 | 3.5 | 4.2 | 3.8 | 10.9 | 21.3 | 19.9 | 22.6 |
| P | Insulin | mIU/L | M+F | All | BMI≥28 | 503 | 1.7 | 2.2 | 2.0 | 4.4 | 10.5 | 9.6 | 11.4 |
| P | Cortisol | nmol/L | M+F | All | | 736 | 151 | 173 | 162 | 337 | 606 | 588 | 624 |
| NP | GH | µg/L | M | All | | 341 | 0.01 | 0.01 | 0.01 | 0.04 | 2.99 | 1.47 | 4.51 |
| NP | | | F | All | | 396 | 0.03 | 0.05 | 0.04 | 0.81 | 7.90 | 6.82 | 8.97 |
| P | PRL | µg/L | M | All | | 340 | 3.2 | 3.8 | 3.5 | 7.5 | 16.3 | 14.9 | 17.7 |
| P | | | F | PreMP | OC, TβhCG ≥2.9 | 242 | 3.7 | 4.9 | 4.3 | 10.8 | 30.0 | 24.6 | 35.3 |
| P | | | F | PostMP | | 118 | 2.7 | 3.5 | 3.1 | 6.5 | 16.1 | 13.7 | 18.4 |
| P | LH | IU/L | M | All | | 336 | 1.16 | 1.63 | 1.39 | 3.17 | 8.12 | 7.20 | 9.029 |
| P | | | F | PreMP | OC, TβhCG ≥2.9 | 241 | 1.65 | 2.39 | 2.02 | 6.70 | 42.5 | 34.6 | 50.4 |
| P | | | F | PostMP | | 117 | 4.2 | 12.6 | 8.4 | 28.4 | 61.1 | 51.7 | 70.6 |
| P | FSH | IU/L | M | <45 | | 203 | 1.12 | 1.44 | 1.28 | 3.52 | 9.5 | 8.2 | 10.8 |
| P | | | M | ≥45 | | 136 | 2.10 | 2.77 | 2.43 | 5.20 | 20.2 | 14.9 | 25.4 |
| P | | | F | PreMP | OC, TβhCG ≥2.9 | 237 | 1.60 | 2.83 | 2.22 | 6.13 | 27.3 | 8.9 | 45.7 |
| P | | | F | PostMP | | 118 | 12 | 30 | 21 | 73 | 138 | 125 | 150 |
| P | TβhCG | IU/L | F | PreMP | OC | 222 | 0.07 | 0.16 | 0.11 | 0.54 | 1.84 | 1.49 | 2.20 |
| P | | | F | PostMP | | 109 | 0.56 | 1.23 | 0.90 | 3.04 | 8.20 | 7.07 | 9.33 |
| P | Estradiol | pmol/L | M | All | | 339 | 2 | 12 | 7 | 72 | 175 | 162 | 188 |
| P | | | F | PreMP | OC, TβhCG ≥2.9 | 245 | 5 | 30 | 17 | 310 | 1519 | 1314 | 1725 |
| NP | | | F | PostMP | | 118 | 4.0 | 4.0 | 4.0 | 29 | 466 | 210 | 1224 |
| P | Progesterone | nmol/L | M | <45 | | 204 | 0.34 | 0.54 | 0.44 | 2.09 | 5.28 | 4.73 | 5.83 |
| P | | | M | ≥45 | | 135 | 0.25 | 0.53 | 0.39 | 1.46 | 4.10 | 3.21 | 4.98 |
| NP | | | F | PreMP | OC, TβhCG ≥2.9 | 245 | 0.13 | 0.55 | 0.34 | 3.67 | 54.9 | 47 | 62 |
| P | | | F | PostMP | | 116 | 0.00 | 0.20 | 0.10 | 0.84 | 3.36 | 2.25 | 4.48 |
| P | Testosterone | nmol/L | M | All | | 338 | 6.5 | 7.2 | 6.9 | 12.3 | 22.5 | 21.4 | 23.7 |
| P | | | F | <45 | OC | 200 | 0.34 | 0.58 | 0.46 | 1.56 | 2.96 | 2.70 | 3.22 |
| P | | | F | ≥45 | OC | 171 | 0.10 | 0.29 | 0.19 | 1.03 | 2.17 | 1.98 | 2.36 |
| P | SHBG | nmol/L | M | All | | 293 | 10 | 13 | 11 | 31 | 74 | 66 | 82 |
| P | PTH | ng/L | M+F | All | | 732 | 18 | 20 | 19 | 39 | 74 | 69 | 78 |
| P | TSH | mIU/L | M+F | All | AIT | 599 | 0.6 | 0.7 | 0.6 | 1.6 | 3.8 | 3.5 | 4.0 |
| P | FT4 | pmol/L | M+F | All | AIT | 598 | 8.1 | 8.7 | 8.4 | 11.1 | 14.2 | 14.0 | 14.5 |
| P | FT3 | pmol/L | M | All | AIT | 220 | 4.21 | 4.50 | 4.35 | 5.25 | 6.15 | 6.00 | 6.30 |
| P | | | F | All | AIT | 211 | 4.04 | 4.23 | 4.14 | 4.88 | 6.09 | 5.92 | 6.27 |
| P | TT4 | nmol/L | M+F | All | AIT | 567 | 64 | 69 | 67 | 93 | 127 | 124 | 131 |
| P | TT3 | nmol/L | M+F | All | AIT | 561 | 1.2 | 1.3 | 1.3 | 1.6 | 2.1 | 2.0 | 2.1 |

P = parametric; NP = nonparametric; LL = lower limit; UL = upper limit; Me = median; CI = confidence interval; OC = oral contraceptives; PreMP = premenopausal; PostMP = postmenopausal; AIT = autoimmune thyroiditis.

RIs for TPOAb and TgAb were not determined with availability of the cutoff values for diagnostic use. The first column stands for distinction between parametric (P) method and nonparametric (NP) method for RI derivation, the 4th and 5th columns for partitioning by sex and age, respectively, and the 6th column for exclusion criteria.

The accuracy of Gaussian transformation by use of the modified Box-Cox formula is shown in S2 Fig. In the comparison between P and NP methods, 90% CI of RI limits were almost invariably narrower and upper limits tended to be higher as was described clearly in the part one of this report (the data omitted with similar tendencies). Exception for this general rule were encountered in deriving RIs for three analytes: GH, progesterone of premenopausal females, estradiol of postmenopausal females. Their RVs failed to attain Gaussian distribution even after power transformation with presence of bimodal peaks (progesterone, and estradiol) or many values below the detection limits (10% of males for GH). Therefore, NP method was used for derivation of their RIs. Among analytes with BMI-related changes (insulin, GH, testosterone and SHBG), the effect of excluding individuals with BMI≥28 was found effective for insulin in lowering UL from 15.7 to 10.5 mIU/L, but not for the other three. $BR_{UL}$ after partition by BMI was −1.49, which far exceeded the critical value of |BR|>0.375 (S4 Table). Therefore, we adopted the BMI restricted RI for insulin.

As described in the Methods, after the completion of data analysis for this study, new assay methods for TSH and TβhCG became available. Therefore, we re-measured remaining serum aliquots from the volunteers stored at −80˚C using the new assays after confirming the stability of the analytes. Method comparison between the old and new reagents was performed using the major-axis linear regression after logarithmic and square root transformation for TSH and TβhCG, respectively. The results are as shown in S3 Fig. Accordingly, the final RIs for TSH and TβhCG listed in Tables 3 and 4 and S4 Table were recalibrated to the values of the new reagents by use of the linear equations.

## Discussion

In this part two report of the Russian RI study, we applied a variety of special techniques required for proper derivation of RIs for a heterogeneous group of immunochemistry tests, consisting of tumor markers, reproductive hormones, thyroid function tests, and miscellaneous hormones. The most important consideration was to properly handle abnormal results attributable to various latent conditions of common occurrence, specific to each analyte. Another important consideration was to carefully explore sex and age-related variations of their RVs to judge the need for partitioning RIs.

### 1) Considerations for abnormal results among the healthy volunteers

We encountered several situations which required special procedures to deal with high prevalence of abnormal results among apparently healthy individuals.

Regarding the influence of nutritional status, RVs of insulin (both sexes), testosterone (male), SHBG (male), and GH (female) were associated with BMI in that order of strength (Fig 2), as have been reported in [19], [20–22] and [23], respectively. But only for insulin, we found that restricting individuals with BMI≥28 kg/m$^2$ was effective in reducing the influence of overnutrition (S4 Table). The UL of the RI for insulin (10.5 mIU/L) became significantly lower than that of the manufacturer (23 mIU/L). It is still lower than that of the IFCC Asian study (11.8 mIU/L) [4], in which the same immunochemistry analyzer UniCel DxI 800 (Beckman Coulter Inc.) was employed and individuals with BMI≥28 were also excluded. However, in our previous report of RIs for chemistry analytes [2], we found it necessary to exclude volunteers with BMI≥28 for nutritional markers such as uric acid, glucose, triglyceride, ALT, and

**Table 4. Comparison of RIs with other studies and manufacturer (Part 1).**

| Analyte | Unit | Age | This Russian study | | | Asian study | | | Chinese study | IFU Beckman Coulter Access reagents | | |
|---|---|---|---|---|---|---|---|---|---|---|---|---|
| | | | M+F | M | F | M+F | M | F | M | M+F | M | F |
| **AFP** | µg/L | All | | | | 1.1–6.5 | 1.2–6.8 | 1.0–6.4 | | 0–9.0 | | |
| | | <45 | 1.0–7.0 | | | | | | | | | |
| | | ≥45 | 1.5–8.7 | | | | | | | | | |
| **CEA** | µg/L | All | | .48–3.84 | | 0.4–4.1 | 0.4–4.4 | 0.4–3.4 | | 0–3.0 | | |
| | | <45 | | | .33–3.32 | | | | | | | |
| | | ≥45 | | | .47–5.19 | | | | | | | |
| **CA19-9** | kIU/L | All | 2.3–29.3 | | | 0.8–30 | 0.8–24.5 | 0.9–33.3 | | 0–35 | | |
| **CA125** | kIU/L | All | | 3.9–27.5 | 4.9–38.7 | | 3.2–16.2 | 4.2–42.4 | | 0–35 | | |
| **CA15-3** | kIU/L | All | 3.8–21.3 | | | 4.0–19.2 | 4.0–18.8 | 3.9–19.3 | | 0–23.5 | | |
| **Insulin** | mIU/L | All | 2.0–10.5 | | | 1.8–11.8 | 2.1–13.5 | 1.9–10.8 | | 1.9–23 | | |
| **Cortisol** | nmol/L | All | 162–606 | | | 45–193 | 51–197 | 41–190 | | 185–624 | | |
| **GH** | µg/L | All | | .01–2.99 | 0.04–7.9 | | | | | | .003–0.97 | .01–3.61 |
| **PRL** | µg/L | All | | 3.5–16.3 | | 4.0–29 | 4.0–21 | 5–33 | 4.2–21.2 | 2.64–13.1 | All | |
| | | PreMP | | | 4.3–30.0 | | | | | | <50 | 3.34–26.7 |
| | | PostPM | | | 3.1–16.1 | | | | | | ≥50 | 2.74–19.6 |
| **LH** | IU/L | All | | 1.4–8.1 | | | 1–7.0 | 1–71 | 1.6–10 | 1.24–8.62 | All | |
| | | | | | | | | | | | follic | 2.12–10.9 |
| | | | | | | | | | | | median | 19.2–103 |
| | | PreMP | | | 2.0–42.5 | | | | | | lutheal | 1.20–12.9 |
| | | PostPM | | | 8.4–61.1 | | | | | | PostMP | 10.9–58.6 |
| **FSH** | IU/L | <45 | | 1.3–9.5 | | | 2–14.0 | 2–173 | 1.9–16.3 | 1.3–19.3 | All | |
| | | ≥45 | | 2.4–20.2 | | | | | | | follic | 3.85–8.78 |
| | | | | | | | | | | | median | 4.54–22.5 |
| | | PreMP | | | 2.2–27.3 | | | | | | lutheal | 1.79–5.12 |
| | | PostPM | | | 21–138 | | | | | | PostMP | 16.7–113 |
| **TβhCG** | IU/L | All | | | | | | | | | 18–40 | 0.2–0.4 |
| | | PreMP | | | 0.10–1.8 | | | | | | ≥ 40 | 1.1–2.9 |
| | | PostPM | | | 0.9–8.2 | | | | | | PostMP | 6.4–10.4 |
| **Estra- diol** | pmol/L | All | | 6.8–175 | | | 66–140 | 50–840 | 4.7–195 | -172 | All | |
| | | | | | | | | | | | follic | 99–448 |
| | | | | | | | | | | | lutheal | 180–1068 |
| | | PreMP | | | 17–1519 | | | | | | median | 349–1590 |
| | | PostPM | | | 4.0–466 | | | | | | PostMP | -147 |
| **Proge- sterone** | nmol/L | <45 | | 0.4–5.3 | | | 0.37–4.48 | .1–66.5 | | 0.4–6.5 | All | |
| | | ≥45 | | 0.4–4.1 | | | | | | | follic | 0.98–4.8 |
| | | PreMP | | | 0.3–55 | | | | | | lutheal | 16.4–59 |
| | | PostPM | | | 0.1–3.4 | | | | | | PostMP | -2.48 |
| **Testo- sterone** | nmol/L | All | | 6.9–22.5 | | | 10.1–28.4 | 0.9–3.5 | 7.2–24.3 | 6.1–27.1 | All | -2.6 |
| | | | | | | | | | | 9.0–28.3* | 18–30 | |
| | | <45 | | | 0.5–3.0 | | | | | 6.9–23.6* | 31–44 | |
| | | ≥45 | | | 0.2–2.2 | | | | | 5.2–23.7* | 45–66 | |
| **SHBG** | nmol/L | All | | 11–74 | | | | | 11.5–66.3 | 13.3–89.5* | 20–50 | |
| | | <45 | | | | | | | | | 20–46 | 18,2–135 |
| | | ≥45 | | | | | | | | | 47–91 | 16.8–125 |

PreMP = premenopausal; PostPM = postmenopausal; IFU = instruction for use;

* See the right for age groups.

CRP in order to reduce substantial gaps between the ULs of their RIs and clinical decision limits of respective analytes. Therefore, we adopted the insulin RI derived after restricting BMI at the same level although the sample size was reduced by 31%.

The effect of cigarette smoking on RVs is well known for CEA [24]. We confirmed the phenomenon as shown in Fig 2. The frequency of individuals with smoking habits was 30% (103/341) in males and 16% (64/396) in females. Therefore, in derivation of the RI for CEA, we excluded the individuals with smoking habit.

Influences of oral contraceptives (OC) on reproductive hormones were all negligible for the derivation of the RIs with the proportion of premenopausal women on OC at 8% (24/274). However, conforming to the study protocol, we adopted the RIs derived after excluding individuals under OC.

For CA19-9, we observed a cluster of extremely low values among the RVs (S1 Fig). They obviously represent individuals with Lewis-antigen negative phenotype, who do not express the CA19-9 antigen. According to the literature, at least 5–10% of the population do not secrete a detectable level of CA19-9 antigen and about 10% in the white population [25]. The prevalence in our cohort was 10.5% (36/341) in males and 11.3% (45/396) in females.

For thyroid function tests, it is essential to exclude individuals with latent autoimmune thyroiditis. The prevalence of AIT by the criteria of TPOAb$\geq$9 IU/L or TgAb$\geq$4 IU/L shown in the kit insert was 10.3% (35/340) in males and 24.5% (96/392) in females among our volunteers (Fig 3). The prevalence seems somewhat higher compared with those reported by other investigators [26]. The effect of excluding individuals with AIT by the criteria was prominent for TSH, slight for TT4, but negligible for FT4, FT3, and TT3 (Fig 3). The results apparently implied that negative feedback mechanism of pituitary thyroid axis works well to keep thyroxine and triiodothyronine level at normal level by increased secretion of TSH.

## 2) Partition of RVs by age and sex

Another important step prior to the calculation of the RIs was to judge the need for partitioning RVs according to age and sex. Although we adopted SDR$\geq$0.4 as its primary guide, we found it necessary to refer to BR and to visually inspect actual differences. Partition by sex was obviously required in every respect for all the reproductive hormones, and for GH and FT3. As for CA125, the higher values in females at reproductive ages are well known [6]. Nevertheless, the SDR$_{sex}$ was calculated as 0.25. We interpret that the finding of weak sex-difference was confounded by age-related reduction in CA125 only in females (S1 Fig). Therefore, we had planned to partition RVs by sex and then by age for females. However, actual bias at LLs and ULs (BR$_{LL}$ and BR$_{UL}$) after partition at age 45 in females was less than 0.375, and thus, the RIs for CA125 was just set for each sex. Among the thyroid function tests, only FT3 showed a relatively high SDR$_{sex}$ of 0.45 with lower values in females. It was consistent with the finding reported in the Asian study [4].

Among tumor markers, age-related increases in RVs (SDR$_{age}\geq$0.4) was observed in both sexes for AFP, in males for CA19-9, and in females for CEA and CA15-3, while age-related reduction was observed for CA125 in females (S1 Fig). These findings were consistent with previous reports [6] and are important in clinical interpretation of their values. Therefore, we partitioned the RVs at age 45 for AFP and CEA (female). However, for CA125, as described above, and for CA19-9 and CA15-3, the actual differences at LL or UL (BR$_{LL}$ and BR$_{UL}$) after the partition were small, and thus, we did not adopt age-specific RIs for them.

Among the reproductive hormones, as well known, marked menopause-related increases were observed for LH, FSH, and T$\beta$hCG in females. The increase in LH and FSH was also observed in males, but less prominent and more gradual in the pattern. The UL of T$\beta$hCG for

the postmenopausal women (8.2 IU/L) was comparable to those, provided by manufacturer (10.4 IU/L). The age-related changes in RIs are not considered for use in the assessment of malignant conditions, although there are reports that demonstrated clinical utility of hCG in risk assessment of trophoblastic diseases, germ cell tumors, etc. [27]. On the other hand, several publications demonstrated increased hCG level in elderly women, which is possibly explained by production of hCG from pituitary [28]. Therefore, we believe that the newly derived age-specific ULs for TβhCG are important to reduce false-positive judgment of postmenopausal women in the assessment of malignant conditions, like choriocarcinoma.

In contrast to those glycoprotein hormones, prominent age-related decrease in RVs ($SDR_{age}$) was observed in females for estradiol (0.98), progesterone (0.63), testosterone (0.47), prolactin (0.57), and in males for FSH (0.52) and progesterone (0.43). A weaker decrease by age was observed in males for LH (0.39), testosterone (0.18) and prolactin (0.16), and none for estradiol. The reports on age-related changes in testosterone in men are mixed: either decrease [29] or unchanged after 40 yo [30]. In our case, no partition by age was done for males because the difference was slight. The final RI for males of all ages was close to that provided by the manufacturer for middle-age group (6.87−23.56 nmol/L for 31−44 years of age).

In males, we observed prominent negative correlation of SHBG with BMI ($r_p$ = −0.45) and prominent positive correlation with age ($r_p$ = 0.42). A similar trend was observed for $SDR_{BMI}$ (0.35) and $SDR_{age}$ (0.54). However, because BMI increases with age, the associations of BMI and age with SHBG counter-balanced with each other. Therefore, the finale RI for SHBG was not partitioned by age with lack of notable between-age subgroup differences. RIs for SHBG partitioned at 45 years of age differed from those provided by the manufacturer without partition by age (Tables 4 and 5, **Parts 1 and 2**).

Among the thyroid function tests, an age-related decrease was noted slightly in RVs of FT3 in males ($r_p$ = −0.21). The similar male predominant finding has been reported [4]. The age-related decrease is regarded as a physiological adaptation to different metabolic needs in the elderly with reduction in anabolic processes and oxygen consumption [31]. However, in terms of $SDR_{age}$, the values are well below 0.4 in both sexes, however, $BR_{LL}$ or $BR_{UL}$ after partition by age were less prominent. Therefore, no partition by age was performed for FT3.

There are multiple reports on age-related increase in serum TSH level, while we did not observe appreciable change in TSH with age ($SDR_{age}$ = 0.17 for males, $SDR_{age}$ = 0.00 for females). To interpret this discrepancy, note that the NHANES III study [32] showed a progressive elevation of TSH occurs after 40 y.o., but after exclusion of individuals with autoantibodies as we did, the age-related increase is only apparent after 60 y.o. Therefore, a narrower age range of our study may account for possible failure of detecting such an increase of TSH in elderly individuals.

### 3) Comparison to commonly used clinical reference limits

It is important to compare our ULs of RIs for tumor markers with clinical reference limits (cutoff values) provided by the manufacturer. For **CA15-3** and **CA19-9**, our ULs are slightly lower than the cutoff values: 21.3 vs. 23.5 kIU/l, and 29.3 vs. 35 kIU/l, respectively. The cutoff value is generally determined by a case-control study as an optimal value to distinguish the disease group from the non-disease group. It is important to note that the non-disease group is supposed to have the same demographic profile of age, ethnicity, etc. as the disease group. Therefore, the cutoff value tends to dissociate from the UL determined from healthy volunteers. From this perspective, it is of practical importance to interpret the UL in consideration of the cutoff value, if any, so as not to increase the false positive rate of tumor diagnosis.

**Table 5. Comparison of RIs with other studies and manufacturer (part 2).**

| | Analytes | TSH | FT4 | FT3 | TT4 | TT3 | PTH |
|---|---|---|---|---|---|---|---|
| | Unit | mIU/L | pmol/L | pmol/L | nmol/L | nmol/L | ng/L |
| | Age | All | All | All | All | All | All |
| This Russian study | M+F | 0.6–3.8 | 8.4–14.2 | | 67–127 | 1.3–2.1 | 19–74 |
| | M | | | 4.4–6.2 | | | |
| | F | | | 4.1–6.1 | | | |
| Asian study | M+F | 0.4–4.0 | 9.2–14.6 | 3.86–5.5 | | | 21–92 |
| | M | 0.4–3.8 | 9.4–14.9 | 4.05–5.9 | | | 21–89 |
| | F | 0.4–3.9 | 9.1–14.2 | 3.8–5.31 | | | 21–97 |
| Chinese studies | M+F | 0.71–4.87 | 11.45–19.3 | 4.01–6.6 | 77–144 | 1.07–2.0 | |
| | M | 0.71–4.5 | 11.7–19.6 | 4.17–6.78 | 78–146 | 1.1–2.1 | |
| | F | 0.78–5.3 | 11.3–18.7 | 3.89–6.2 | 76–141 | 1.05–1.9 | |
| Italy | M+F | 0.4–3.7 | | | | | |
| | M | | 7.7–13.7 | | | | |
| | F | | 6.8–12 | | | | |
| France | M+F | 0.4–3.6 | | | | | |
| | M | | 9.3–15.1 | | | | |
| | F | | 8.6–14.7 | | | | |
| Germany | M+F | 0.3–3.1 | | | | | |
| | M | | 7.9–13.5 | | | | |
| | F | | 7.3–12.9 | | | | |
| IFU* | M+F | 0.38–5.33 | 7.86–14.41 | 3.8–6.0 | 78.38–157.4 | 1.34–2.73 | 12–88 |

IFU*- instruction for use of Beckman Coulter reagents

In contrast, our UL of **CA125 for females** (38.7 kIU/L) was higher than commonly used cutoff of 35 kIU/L [33]. This difference may be attributable to a high prevalence of endometriosis and other inflammatory gynecological diseases of non-cancerous etiology in Russia. In fact, the incidence of endometriosis increased by 72.9% from 1999 to 2011 [34] after widespread use of CA125 testing. However, it was not possible for us to exclude latent endometriosis with unavailability of relevant information in the questionnaire. In any case, we found that the UL for CA125 reported in the Asian study using the same reagent [4] was quite comparable with our result (Table 4).

Due to between-assay variations, the cutoff value for low testosterone is different depending on studies and societies. The Endocrine Society and the American Urology Association (AUA) recommend using a **total testosterone** <300 ng/dL (10.4 nmol/L) with repeated measurements of morning total testosterone as a reasonable cutoff in support of the diagnosis of low testosterone, preferably using the same laboratory with the same method/instrumentation for measurements. The International Society for the Study of the Aging Male (ISSAM) and the International Society for Sexual Medicine (ISSM) used the cutoff value of total testosterone <12 nmol/L (350 ng/dL). However in 2015, they suggested that testosterone replacement therapy (TRT) may be reasonably offered to symptomatic patients with total testosterone concentration even higher than 12 nmol/L based on clinical judgement [35], which is still far higher than the LL of RI (6.9 nmol/L) for males derived in the present study using the BC analyzer. This discrepancy points to the unstandardized status of the testosterone assay and the need for reagent-specific cutoff value.

For TSH, the American Association of Clinical Endocrinologists (AACE) recommends using a TSH range of 0.3 to 3.0 mIU/L for therapeutic decision since 2003 [36], the European

Thyroid Association (ETA) suggests the reference interval for serum TSH in the general adult population between 0.4 and 4.0 mIU/L [37] and the National Academy of Clinical Biochemistry reported that: "In the future, it is likely that the upper limit of the serum TSH euthyroid reference interval will be reduced to 2.5 mIU/L because 95% of rigorously screened normal euthyroid volunteers have serum TSH values between 0.4 and 2.5 mIU/L" [38]. On the other hand, the RI for TSH derived in this study after exclusion of cases with apparent AIT was 0.6 −3.8 mIU/L. It matches well with those reported in other studies: the median (LL−UL) by a French group were 1.4 (0.4–3.6) mIU/L, by a German group 1.1 (0.3–3.1) mIU/L, by a Italian group 1.4 (0.4–3.7) mIU/L [39]. The RI of this study was shifted to a much lower side from that of the manufacturer (0.38−5.33 mIU/L; Access TSH (3rd IS) (Table 5). In any case, it should be noted, that there is no common RI and the fluctuation of UL range could make from 2.5 to 5.5 mIU/ml. It is apparent that, although CDLs have been proposed by academic societies, they are not generally applicable with apparent lack of harmonization of the TSH assays. In fact, the C-RIDL's interim report on the global RI study clearly showed that after aligning TSH test results based on the commonly tested serum panel, no obvious between-country difference was observed among six countries examined [18]. Therefore, the observed differences among the RIs or CDLs appear not due to ethnic difference, but to non-harmonized test results. This situation of a large between-reagent differences in test results of TSH were clearly documented in the 2017 report of IFCC Committee for Standardization of Thyroid Function Tests [40].

## 4) Comparison of Russian RVs with those of other countries

We compared our RIs or RVs with those of the countries collaborating in the IFCC Asian and global projects [4, 18], those of other relevant studies as well as RIs provided by the manufacturer. We noted several features as follows.

For insulin, after applying exclusion of BMI≥28, the UL of the Russian RI (M+F) became significantly lower compared with that of the manufacturer (10.5 vs. 23 mIU/L), and comparable to Asian study (M:13.5, F:10.8 mIU/L) [4], which employed the same immunoassay analyzer UniCel DxI 800 (Beckman Coulter Inc.) and also applied exclusion of individuals with BMI≥28. However, in reference to C-RIDL' report on the global study (S2 Fig of [18]), the median Russian RVs for insulin was higher than other countries, implying that current manufacturer's RI is set way-higher for appropriate clinical use.

For testosterone, the RI for males derived in this study shifted to a lower side compared to the RIs published in the Asian study [4], and RVs were lower than those of the U.S. and Japan in the interim report of the global study [18]. It should be also noted that the RI by this study is narrow with its UL lower than that shown in the reagent instruction, which was derived based on the U. S. population. However, in consideration of a relatively small SDR for between-country differences for testosterone shown in the global study report [18], our RI seems not biased much.

For TSH, Russia RVs are comparable to those countries that collaborated in the global project [18]. Our RI is also close to those reported in the Asian study and common Europe investigation [4, 39]. At the same time, UL for TSH was higher in a nationwide Chinese study [11] where RIs were also divided by sex (Table 5). The reason is obviously by use of different exclusion criteria for the volunteers.

PTH in males and females in Russia was comparable with other countries, such as Saudi Arabia, Turkey, and U.S., but significantly higher than Pakistan and Philippines. (S2 Fig of [19]). In Asian and current studies, UL for PTH were comparable [4] (Table 5). The between-country difference in RVs of PTH was one of the most significant ones among the analytes examined in both sexes (between-country SDR of 0.63 for male and 0.64 for female) [18].

Cortisol exhibits a slight between-country difference (SDR of 0.28 for male, 0.29 for female) [18]. Median RVs of cortisol in Russian females is close to the U.S. and higher than in Asian countries, India and Saudi Arabia. In males, the RVs are the highest among the countries in the global study. The Russian RI for cortisol is close to that of the manufacturer, but three times higher than that published in the Asian study (Table 4). We do not know whether the higher cortisol level in Caucasians in the U.S. and Russia points to more stress than other countries.

AFP, CEA, CA125, and PRL didn't show significant between-country differences according to the global paper results (SDR of 0.12, 0.13, 0.21 and 0.12 for males and 0.18, 0.15, 0.05 and 0.14 for females, respectively) [18]. For CA-125, the UL for females in the Asian study was higher than those provided in the Russian study and both were higher than UL provided by the manufacturer in the instruction for use. At the same time, the UL for prolactin in males in the Russian study was twice lower than in the Asian study.

Median of LH, FSH were shifted to the right in Turkey and Japan accordingly, but low and upper values of the dispersion were fully comparable (SDR = 0.29 and 0.22). For females, no country differences were observed.

Progesterone had significant between-country differences in males (SDR = 0.91), Russia was higher than other countries, while in females such differences were not observed (SDR 0.1) (Table 4).

## Conclusion

This is the first comprehensive Russian study for derivation of RIs for 22 major immunochemistry parameters consisting of tumor markers, thyroid function tests, vitamins, reproductive and other hormones. The study was conducted by use of the internationally harmonized protocol elaborated by C-RIDL, IFCC with recruitment of 758 well-defined, apparently healthy adults from three major cities in Russia.

No regional differences among the three major cities were observed in any parameter. Careful assessment and exclusion of latent abnormal values of common occurrence was a crucial step. Close associations of BMI with RVs were observed for insulin, testosterone (M), SHGB (M), and GH (F) in that order of strength. For insulin, exclusion of individuals with BMI$\geq$28 was effective in lowering the UL of RI, but not for others. In the derivation of the RI for CA19-9, individuals with apparent Lewis-negative blood type (M: 10.5%, F: 11.3%) were excluded. For thyroid function tests, individuals with AIT (M:10.3%, F: 24.5%) were excluded, but the procedure only affected the RI for TSH. Partition of RVs by sex was required for all reproductive hormones, CA125, CEA, and GH. Partition by age was required for AFP, for CEA (F), and for all reproductive hormones (F).

A majority of RIs derived in this study differed from those provided by the manufacturers. This fact points to the importance of establishing the country specific RIs. A variable degrees of differences were noted from CDLs (or cutoff values) set by clinical guidelines for CA19-9, CA125, testosterone, insulin, and TSH. Although some of the differences are attributable to the lack of harmonization in test results, they are inevitable from theconcept of the RI as "health"-associated range, which is distinct from the concept of cutoff value that requires the case-control study for its determination.

## Supporting information

**S1 Dataset.**
(XLSX)

**S1 Fig. Sex and age-related changes in RVs of all immunoassay analytes.** Distributions of RVs for all the analytes were shown after subgrouped by sex and age. No secondary exclusion was performed in plotting data. The box in each scattergram represents central 50% range and the vertical bar in the middle represents a median point. On top of each scattergram, the magnitudes of between-sex and between-age variations are shown as $SDR_{sex}$ and $SDR_{age}$ derived separately for males (M) and females (F).
(PDF)

**S2 Fig. Accuracy of power transformation used in the parametric method.** RIs were derived by both parametric and nonparametric methods. The accuracy of Gaussian transformation by Box-Cox formula can be assessed from theoretical Gaussian curves in two histograms shown on left top (before and after the transformation). The results of by Kolmogorov-Smirmov (K-S) test for normality of distribution were shown on right upper panel. The accuracy of the transformation can be also seen from the linearity in the probability paper plot on the right. The limits of the RI by nonparametric method corresponds to the points where red zigzag line intersect with horizontal 2.5 and 97.5% red lines of cumulative frequencies.
(PDF)

**S3 Fig. Comparison of test results for TSH and TβhCG before and after reagent changes.** Aliquots of volunteers' sera stored at −80C˚ were tested in 2018 by use of new reagents for TSH and TβhCG after confirmation of the stability of the analytes. Recalibration of values by the old reagent was performed using the major-axis linear regression between new and old values after logarithmic and square-root transformation for TSH and TβhCG, respectively.
(PDF)

**S1 Table. Demographic profile of volunteers.**
(PDF)

**S2 Table. Characteristics of assays for analytes examined in this study.**
(PDF)

**S3 Table. Comparison of assay characteristics for TSH and TβhCG before and after reagent changes.**
(PDF)

**S4 Table. List of RIs derived in various conditions partitioned by sex and/or age (menopausal state) with/without secondary exclusion options.**
(PDF)

## Acknowledgments

The authors express sincere gratitude to Beckman Coulter, LLC (Russia) for their generous support of the assay reagents. We are grateful to Yury Andreychuk, CEO of Helix Laboratories Services, and the Helix staff for their kind assistance in the recruitment of volunteers, sample preparations, and provision of sampling equipment.

## Author Contributions

**Conceptualization:** Kiyoshi Ichihara.

**Data curation:** Anna Ruzhanskaya, Kiyoshi Ichihara, Svetlana Evgina, Irina Skibo, Nina Vybornova, Anton Vasiliev, Galina Agarkova, Vladimir Emanuel.

**Formal analysis:** Anna Ruzhanskaya, Kiyoshi Ichihara, Galina Agarkova.

**Funding acquisition:** Anna Ruzhanskaya, Irina Skibo.

**Investigation:** Anna Ruzhanskaya, Kiyoshi Ichihara, Nina Vybornova, Anton Vasiliev, Galina Agarkova.

**Methodology:** Anna Ruzhanskaya, Kiyoshi Ichihara, Galina Agarkova.

**Project administration:** Anna Ruzhanskaya, Kiyoshi Ichihara, Svetlana Evgina, Irina Skibo, Nina Vybornova, Anton Vasiliev, Vladimir Emanuel.

**Resources:** Anna Ruzhanskaya, Irina Skibo, Anton Vasiliev.

**Software:** Kiyoshi Ichihara.

**Supervision:** Anna Ruzhanskaya, Kiyoshi Ichihara, Svetlana Evgina, Irina Skibo, Nina Vybornova.

**Validation:** Anna Ruzhanskaya, Kiyoshi Ichihara, Galina Agarkova.

**Visualization:** Kiyoshi Ichihara, Anton Vasiliev.

**Writing – original draft:** Anna Ruzhanskaya, Kiyoshi Ichihara.

**Writing – review & editing:** Anna Ruzhanskaya, Kiyoshi Ichihara.

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
