## [Decision Letter · Decision Letter 0]

8 Oct 2020

PONE-D-20-15086

Sources of variation and establishment of Russian reference intervals for major hormones and tumor markers

PLOS ONE

Dear Dr. Ruzhanskaya,

Thank you for submitting your manuscript to PLOS ONE. After careful consideration, we feel that it has merit but does not fully meet PLOS ONE’s publication criteria as it currently stands. Therefore, we invite you to submit a revised version of the manuscript that addresses the points raised during the review process.

Specifically, the reviewers thought that a more detailed materials section, including a description of the percentages of male and females in the study, as well as a better rational for excluding patients with BMI≥28 would improve the manuscript. Please submit a revised manuscript addressing the concerns by the reviewers by Nov 22 2020 11:59PM.

If you will need more time than this to complete your revisions, please reply to this message or contact the journal office at plosone@plos.org. Please include the following items when submitting your revised manuscript:

We look forward to receiving your revised manuscript.

Kind regards,

Colin Johnson, Ph.D.

Academic Editor

PLOS ONE

Journal Requirements:

4. Thank you for stating the following in the Financial Disclosure section:

"The author(s) received no specific funding for this work"

We note that one or more of the authors are employed by a commercial company: "Beckman Coulter LLC and Helix Laboratory"

5. Thank you for including your ethics statement on the online submission form: 

"The study protocol, including methods for invitation, provision of information about the

study for volunteers, taking informed consent and questionnaire regarding current

health status and lifestyle, were written approved by the Ethic Committee of City

Hospital #40, Saint-Petersburg.".  

To help ensure that the wording of your manuscript is suitable for publication, would you please also add this statement at the beginning of the Methods section of your manuscript file.

Reviewers' comments:

Reviewer's Responses to Questions

**Comments to the Author**

1. Is the manuscript technically sound, and do the data support the conclusions?

Reviewer #1: Yes

Reviewer #2: Yes

2. Has the statistical analysis been performed appropriately and rigorously? 

Reviewer #1: Yes

Reviewer #2: Yes

3. Have the authors made all data underlying the findings in their manuscript fully available?

Reviewer #1: Yes

Reviewer #2: Yes

4. Is the manuscript presented in an intelligible fashion and written in standard English?

Reviewer #1: Yes

Reviewer #2: Yes

5. Review Comments to the Author

Reviewer #1: The present authors have set reference intervals (RI) for 24 immunochemical analytes, consisting of clinically relevant tumor markers and hormones, in Russian population. It is based on an initial study on RI of routine biochemical items, more than two thirds of which are internationally standardized. Present analytes have been non-standardized or mostly non-harmonized yet. Sorting out healthy reference individuals carefully, they have challenged to yield coherent RI by specifying source of variations (SV) of reference values (RV), while eliminating non-specific, bias-causing factors. Precise systemic studies, covering wide ranges of theirs, have been barely performed so far in relevance with age, sex, lifestyle, and BMI. Their invented, statistical approaches have newly found influencing factors of variable degrees in each item. Russian specific matters are well delineated in discussion. Further development of standardization or harmonization is awaited, nonetheless the RI along with SV presented here is the referable landmark in every clinical setting and can be shared globally beyond one country.

It will also lead to in-depth physiologic and pathophysiologic insights on what is occurring in the body.

Paper is suitable in inclusion in the journal after checking minor points shown below.

Page 6 in BRLL and BRUL formula

Explain where 3.92 is derived.

Page 7 line 2 in sources of variation of RVs

Check the number of parameters: 23, not 24?

Page 7 Line 5 in sources of variation of RVs

TgAb is not included in RI in thyroid function tests. Is it excluded?

Page 13 Results of H-beta-hCG on new reagent related to Supplement Fig. 3

Some apparent discrepancies are observed in the low level of H-beta-hCG between new and conventional reagent. How degree did the result cause its RI change ? It would be shown if the difference is of significant

presence between the two.

Page 15 in the last paragraph

Age-related elevation of SHBG, in which MW is 40000, can be explained by decrease of GFR as physiologic

aging process. The finding is common feature of protein with a MW less than 50000. If it has statistical

association with eGFR or creatinine, involvement in Discussion is preferred.

Page 18-19 Adding discussion on TSH RI

The paper would be more attracting if the authors comparatively discuss the RI of 0.56-4.27 mIU/L in US

according to current successful harmonization project on TSH by IFCC-STFT, that appeared in Clin Chem

2017; 63(7): 1248-1260.

Reviewer #2: Editor in Chief, PLOS ONE

Sources of variation and establishment of Russian reference intervals for major

Hormones and tumor markers

PONE-D-20-15086

Reviewer(s)' Comments to Author:

Reviewer:

Comments to the Author

The manuscript describes the reference interval for most common immunoassays and discuss the sources of variation. The study is part of the global reference interval study of the IFCC.

The practical message of the manuscript is relevant and the statistical elaboration of data is very detailed. Some core aspects have weakened the manuscript.

Specific comments

Manuscript:

Introduction

- Page 4 Line 5-14: the listed information in this paragraph is repeated under materials and methods. The author can describe briefly the main outcomes of the previous study only.

Material and methods

Source data and target analytes

- Page 5: What are the males and females percentages in each region?

- Source and anthropometric data to be described by creating a new table (total and categorized subjects distribution between different regions, males and female counts, sex percentages, BP, BMI etc.).

- All described tests involved in the study can be summarized by creating a new table showing: full name of each test, its abbreviation, unit, and principle of measurement.

- How the new assays for both TSH and TβhCG are different from the old assays?

i.e. Are they new generation, new lots, new components or antibodies…? etc.

Quality Control

- Page 5: The CV value for each test and the allowable limits described in the Westgard Website are to be listed in the previous suggested new table which contain tests names.

Results:

Miscellaneous hormones

- Page 11: excluding subjects with BMI≥28 is not clearly justified. It is well known that insulin is highly associated with BMI. Therefore, it is not convincing that the UL insulin RI for Russians is smaller than Asians with lower BMI (page14). In the first paper for Russian chemistry analytes RIs the following was described: “The proportion of recruited obese volunteers with BMI>28 was 34% in males (127 persons) and 30% in females (125 persons)”. Therefore, these percentages are not small enough to be ignored in the statistical analysis of the derived RI for insulin.

The justification needs to be discussed under the light of BMI prevalence in Russia (1st paragraph page 14) and to be supported by the following data in the section of results:

a- Insulin RI categorized by the BMI

b- Insulin RI calculated by NP method

c- Insulin RI without excluding subjects with BMI≥28.

- Page 11:

Derivation of RIs:

It seems the written paragraph under this title (last paragraph page 11) is truncated.

“The first column stands for distinction between parametric (P) method and nonparametric (NP) method for RI derivation, the fourth column for……?”

- Page 13 Line 4-10: this paragraph is not clearly understood. It needs to be reworded.

- Table 4 (Part 1): the column of Russian study to be corrected to Current Russian study.

- Table 4 (Part 2): Chinese studies*. What does the asterisk mean?

- Page 17:

Distinction between RIs and CDLs

“For CA15-3 and CA19-9, our ULs are lower than commonly used cutoff values: 21.3 vs. 23.5KIU/l, and 29.3 vs. 35 KIU/l, respectively. The value between the ULs and cutoff values may be regarded as a gray zone for early detection of adenocarcinoma, although an increased false-positive rate is a problem in prioritizing the ULs.”

The author needs to clarify what is the point of getting RIs for Tumor Markers as long as the rate of false-positive results will be increased if ULs is adapted.

- Page 17: The full names for the abbreviation ISSAM, ISSM and TRT are required.

6. PLOS authors have the option to publish the peer review history of their article (what does this mean?). If published, this will include your full peer review and any attached files.

Reviewer #1: No

Reviewer #2: No

---

## [Author Response · Author response to Decision Letter 0]

11 Nov 2020

Our answers to the requests from the reviewers as below:

Reviewer #1: 

The present authors have set reference intervals (RI) for 24 immunochemical analytes, consisting of clinically relevant tumor markers and hormones, in Russian population. It is based on an initial study on RI of routine biochemical items, more than two thirds of which are internationally standardized. Present analytes have been non-standardized or mostly non-harmonized yet. Sorting out healthy reference individuals carefully, they have challenged to yield coherent RI by specifying source of variations (SV) of reference values (RV), while eliminating non-specific, bias-causing factors. Precise systemic studies, covering wide ranges of theirs, have been barely performed so far in relevance with age, sex, lifestyle, and BMI. Their invented, statistical approaches have newly found influencing factors of variable degrees in each item. Russian specific matters are well delineated in discussion. Further development of standardization or harmonization is awaited, nonetheless the RI along with SV presented here is the referable landmark in every clinical setting and can be shared globally beyond one country.

It will also lead to in-depth physiologic and pathophysiologic insights on what is occurring in the body.

Paper is suitable in inclusion in the journal after checking minor points shown below.

Our response→ We are very grateful for your appreciation of our work, and for the kind offer of invaluable comments, which helped us to improve the manuscript. We have carefully gone through them and responded to each one by one. 

Page 6 in BRLL and BRUL formula

Explain where 3.92 is derived.

Our response→ The denominator of the formula represents a standard deviation (SD) comprising the RI. Because LL and UL (lower and upper limits of the RI) can be calculated as LL = mean – 1.96 SD and UL = mean + 1.96 SD, SD can be obtained by solving the two formula as SD = (UL–LL) /3.92. In other words, between LL and UL, there is 3.92 times of SD. We explain this point in the revised manuscript as follow: “The denominator of each formula represents the standard deviation (SDRI) comprising the RI, the width of which corresponds to 3.92 times SDRI.”

Page 7 line 2 in sources of variation of RVs

Check the number of parameters: 23, not 24?

Our response→ Thank you for pointing out the problem. The number of parameters we analyzed was 24. However, we did not determine RIs for TPOAb and TgAb, therefore, the number of analytes that were served for the determination was 22. 

Page 7 Line 5 in sources of variation of RVs

TgAb is not included in RI in thyroid function tests. Is it excluded?

Our response→ Yes, we chose not to determine RIs for TgAb as well as for TPOAb because of presence of many subjects with autoimmune thyroiditis. [Please note that it was a mistake that we included TPOAb results in Table 1 and 2]. In this study, we just used the cutoff values of TPOAb and TgAb as exclusion criteria for derivation of RIs for the parameters in the thyroid panel. We have described the use of those cutoff values in the Methods as “We first identified cases with subclinical autoimmune thyroiditis (AIT) by use of the criteria of TgAb≥4 KIU/L or TPOAb≥9 KIU/L, which are provided in the kit inserts. Prior to deriving RIs, we found the cutoff values were appropriate as a proximal point of tailing values in the distributions in Suppl. Fig 1 (two figures in the last page)” 

Page 13 Results of H-beta-hCG on new reagent related to Supplement Fig. 3

Some apparent discrepancies are observed in the low level of H-beta-hCG between new and conventional reagent. How degree did the result cause its RI change ? It would be shown if the difference is of significant presence between the two.

Our response→ Thanks for raising the important point. We presented the RI for total βhCG measured by the new reagent as 0.11~1.84 (preMP), 0.90~8.20 (postMP) IU/L, which correspond to 0.44~2.27 (preMP), 1.36~7.585 (postMP) IU/L by the old reagent. In the revised Suppl Fig 3, we presented the RIs by each of the two reagents.

Page 15 in the last paragraph

Age-related elevation of SHBG, in which MW is 40000, can be explained by decrease of GFR as physiologic aging process. The finding is common feature of protein with a MW less than 50000. If it has statistical association with eGFR or creatinine, involvement in Discussion is preferred.

Our response→ According to the suggestion, we evaluated the correlations between SHBG and Cre or eGFR. The results are as below. We could not detect appreciable dependency of SHBG by the renal function.

Page 18-19 Adding discussion on TSH RI

The paper would be more attracting if the authors comparatively discuss the RI of 0.56-4.27 mIU/L in US according to current successful harmonization project on TSH by IFCC-STFT, that appeared in Clin Chem 2017; 63(7): 1248-1260.

Our response→ Thank you so much for sharing the paper. We found the paper clearly documented the presence of a large between-reagent differences in TSH measurements and verified the harmonizability of all the assays by implementing a consensus standard. To our disappointment, no globally applicable RI was provided in consideration of possible age and ethnicity differences in the RI. Therefore, we just cited the paper as “This situation of a large between reagent differences in test results of TSH were clearly documented in the 2017 report of IFCC Committee for Standardization of Thyroid Function Tests [40].”

Reviewer #2: Comments to the Author

The manuscript describes the reference interval for most common immunoassays and discuss the sources of variation. The study is part of the global reference interval study of the IFCC.

The practical message of the manuscript is relevant, and the statistical elaboration of data is very detailed. Some core aspects have weakened the manuscript.

Our response→ We are very grateful to you for critically reviewing our manuscript and offering us invaluable comments to improve the manuscript. We addressed each issue diligently. Although some of them are challenging, we answer to each honestly one by one as bellow. 

1/ Introduction

- Page 4 Line 5-14: the listed information in this paragraph is repeated under materials and methods. The author can describe briefly the main outcomes of the previous study only.

Our response→ We appreciate your comment and reduced the redundant texts accordingly.

2/Material and methods

Source data and target analytes

2.1 Page 5: What are the males and females percentages in each region?

Our response→ We prepared a new table (Suppl Table 1) to show the demography as requested.

2.2- Source and anthropometric data to be described by creating a new table (total and categorized subjects distribution between different regions, males and female counts, sex percentages, BP, BMI etc.).

Our response→ As mentioned above, we added Suppl Table 1 that shows detailed demographic information, except for BP that we did not measure.

2.3- All described tests involved in the study can be summarized by creating a new table showing: full name of each test, its abbreviation, unit, and principle of measurement.

Our response→ Thank you for this constructive comment. We newly made Suppl Table 2 that contains all the detailed analytical information. 

2.4- How the new assays for both TSH and TβhCG are different from the old assays?

i.e. Are they new generation, new lots, new components or antibodies…? etc.

Our response→ According to your important suggestion, we newly prepared Suppl Table 3 to clarify the differences between the old and new assay method for the two analytes.

3/ Quality Control

- Page 5: The CV value for each test and the allowable limits described in the Westgard Website are to be listed in the previous suggested new table which contain tests names.

Our response→ Thank you for your comment; this information certainly is very important. We have created and added it in the Suppl Table 2. We would like just to suggest to provide the information from EFLM website instead of the one from Westgard’ website, because according to our information it updates regularly and contains confidence intervals for mean CV of biological variation. 

4/ Results: 

4.1 Miscellaneous hormones 

- Page 11: excluding subjects with BMI≥28 is not clearly justified. It is well known that insulin is highly associated with BMI. Therefore, it is not convincing that the UL insulin RI for Russians is smaller than Asians with lower BMI (page14). In the first paper for Russian chemistry analytes RIs the following was described: “The proportion of recruited obese volunteers with BMI>28 was 34% in males (127 persons) and 30% in females (125 persons)”. Therefore, these percentages are not small enough to be ignored in the statistical analysis of the derived RI for insulin. 

The justification needs to be discussed under the light of BMI prevalence in Russia (1st paragraph page 14) and to be supported by the following data in the section of results: 

a- Insulin RI categorized by the BMI 

b- Insulin RI calculated by NP method 

c- Insulin RI without excluding subjects with BMI≥28. 

Our response→ We appreciate for pointing out the important issue on the BMI restriction in determining the RI for insulin. Regarding how much difference occurs in the RI with/without excluding individuals with BMI≥28 kg/m2, we had described in the Results. However, we added the following texts to note the magnitude of the difference “BRUL after partition by BMI was −1.49, which far exceeded the critical value of |BR|>0.375 (Suppl. Table 4). Therefore, we adopted the BMI restricted RI.”. Regarding the RI by the NP method, we observed no difference from the RI by the parametric method as we showed it Suppl Fig. 2. Therefore, we regarded the NP-RI irrelevant to the readers and withheld it in the Results.

As for the justification of excluding subjects with BMI≥28 even though our UL became lower than that of Asian, it is based on our previous experience in determining chemistry RIs. We found prominent gaps between the RI-ULs and clinical decision limits (CDLs) for nutritional markers such as glucose, uric acid, triglyceride, etc. Therefore, we found it necessary to reduce the gaps by applying the BMI restriction at the level of 28. The finding clearly pointed out that individuals with BMI>28 were obviously not healthy and thus not appropriate for determining the RIs for the nutritional markers. Because insulin was found prominently affected by obesity, we were obliged to use the same cutoff value of BMI as the exclusion criteria for determining the nutritional maker RIs, even though the sample size was reduced. Whereas we could not explain why the adjusted UL was a little lower than Asian: i.e., maybe unharmonized status of the assay. 

As an amendment, we added the following descriptions, at the first subheading of the Discussion (1.Considerations for abnormal results among the healthy volunteers), as to why we adopted the insulin RI after excluding subjects with BMI≥28: 

“The UL of the RI for insulin (10.5 mIU/L) became significantly lower than that of the manufacturer (23 mIU/L). It is still lower than that of the IFCC Asian study (11.8 mIU/L) [4], in which the same immunochemistry analyzer UniCel DxI 800 (Beckman Coulter Inc.) was employed and individuals with BMI>28 were also excluded. However, in our previous report of RIs for chemistry analytes [2], we found it necessary to exclude volunteers with BMI>28 for nutritional markers such as uric acid, glucose, triglyceride, ALT, and CRP in order to reduce substantial gaps between the ULs of their RIs and clinical decision limits of respective analytes. Therefore, we adopted the insulin RI derived after restricting BMI at the same level although the sample size was reduced by 31%.”. 

4.2- Page 11: 

Derivation of RIs: 

It seems the written paragraph under this title (last paragraph page 11) is truncated. 

“The first column stands for distinction between parametric (P) method and nonparametric (NP) method for RI derivation, the fourth column for……?” 

Our response→ Thank you for your comment, we apologize for the careless typos, the phrase was changed.

4.3 - Page 13 Line 4-10: this paragraph is not clearly understood. It needs to be reworded. 

- Table 4 (Part 1): the column of Russian study to be corrected to Current Russian study. 

Our response→ Thank you, the formulation was corrected 

4.4 - Table 4 (Part 2): Chinese studies*. What does the asterisk mean? 

Our response→ We apologize for the careless typos, the asterisk was removed.

- Page 17: 

4.5 Distinction between RIs and CDLs 

“For CA15-3 and CA19-9, our ULs are lower than commonly used cutoff values: 21.3 vs. 23.5KIU/l, and 29.3 vs. 35 KIU/l, respectively. The value between the ULs and cutoff values may be regarded as a gray zone for early detection of adenocarcinoma, although an increased false-positive rate is a problem in prioritizing the ULs.” 

The author needs to clarify what is the point of getting RIs for Tumor Markers as long as the rate of false-positive results will be increased if ULs is adapted. 

Our response→ Thank you for raising the important issue. We first changed the term CDLs in the sub-heading to “clinical reference limits” that imply both the cutoff value and the clinical decision limit for general chemistry analytes. We modified the above sentence as follow to distinguish the UL of the RI from the cutoff value set for tumor makers and hormones for diagnosing specific diseases. “For CA15-3 and CA19-9, our ULs are slightly lower than the cutoff values: 21.3 vs. 23.5kIU/l, and 29.3 vs. 35 kIU/l, respectively. The cutoff value is generally determined by a case-control study as an optimal value to distinguish the disease group from the non-disease group. It is important to note that non-disease group are supposed to have the same demographic profile of age, ethnicity, etc. as the disease group. Therefore, the cutoff value tends to dissociate from the UL determined from healthy volunteers, and thus it is of practical importance to interpret the UL in consideration of the cutoff value, if any, so as not to increase the false positive rate of tumor diagnosis.”

4.6 - Page 17: The full names for the abbreviation ISSAM, ISSM and TRT are required. 

Our response→ Thank you for your comment, we add the full names at the section of the abbreviation list at the top of the manuscript.

---

## [Decision Letter · Decision Letter 1]

17 Nov 2020

Sources of variation and establishment of Russian reference intervals for major hormones and tumor markers

PONE-D-20-15086R1

Dear Dr. Ruzhanskaya,

We’re pleased to inform you that your manuscript has been judged scientifically suitable for publication and will be formally accepted for publication once it meets all outstanding technical requirements.

Kind regards,

Colin Johnson, Ph.D.

Academic Editor

PLOS ONE

Additional Editor Comments (optional):

Reviewers' comments:

Reviewer's Responses to Questions

**Comments to the Author**

1. If the authors have adequately addressed your comments raised in a previous round of review and you feel that this manuscript is now acceptable for publication, you may indicate that here to bypass the “Comments to the Author” section, enter your conflict of interest statement in the “Confidential to Editor” section, and submit your "Accept" recommendation.

Reviewer #1: All comments have been addressed

Reviewer #2: All comments have been addressed

2. Is the manuscript technically sound, and do the data support the conclusions?

Reviewer #1: Yes

Reviewer #2: Yes

3. Has the statistical analysis been performed appropriately and rigorously? 

Reviewer #1: Yes

Reviewer #2: Yes

4. Have the authors made all data underlying the findings in their manuscript fully available?

Reviewer #1: Yes

Reviewer #2: Yes

5. Is the manuscript presented in an intelligible fashion and written in standard English?

Reviewer #1: Yes

Reviewer #2: Yes

6. Review Comments to the Author

Reviewer #1: In response to all requests, the authors have properly answered them. The paper is worth publishing in the journal. It will provide a platform for implementation of every day's clinical practices as well as clinical studies in specified area like endocrines and classical tumour markers.

Reviewer #2: All answers have been addressed as requested by the reviewer.

The reviewer is satisfied with all revised texts, tables and figures.

7. PLOS authors have the option to publish the peer review history of their article (what does this mean?). If published, this will include your full peer review and any attached files.

Reviewer #1: No

Reviewer #2: **Yes: **Anwar Borai

---

## [Editor Report · Acceptance letter]

14 Dec 2020

PONE-D-20-15086R1 

Sources of variation and establishment of Russian reference intervals for major hormones and tumor markers. 

Dear Dr. Ruzhanskaya:

I'm pleased to inform you that your manuscript has been deemed suitable for publication in PLOS ONE. Congratulations! Your manuscript is now with our production department. 

Kind regards, 

on behalf of

Dr. Colin Johnson 

Academic Editor

PLOS ONE